# Effects of Methods and Durations of Extraction on Total Flavonoid and Phenolic Contents and Antioxidant Activity of Java Cardamom (*Amomum compactum* Soland Ex Maton) Fruit

**DOI:** 10.3390/plants11172221

**Published:** 2022-08-27

**Authors:** Waras Nurcholis, Rahma Alfadzrin, Nurul Izzati, Rini Arianti, Boglárka Ágnes Vinnai, Fadillah Sabri, Endre Kristóf, I Made Artika

**Affiliations:** 1Tropical Biopharmaca Research Center, IPB University, Bogor 16151, Indonesia; 2Department of Biochemistry, Faculty of Mathematics and Natural Sciences, IPB University, Bogor 16680, Indonesia; 3Department of Biochemistry and Molecular Biology, Faculty of Medicine, University of Debrecen, H-4032 Debrecen, Hungary; 4Doctoral School of Molecular Cell and Immune Biology, University of Debrecen, H-4032 Debrecen, Hungary; 5Department of Physical Education, Healthy, and Recreation, Faculty of Teacher Training and Education, Muhammadiyah University Bangka Belitung, Pangkalanbaru 33684, Indonesia; 6Department of Civil Engineering, Faculty of Technique and Sciences, Muhammadiyah University Bangka Belitung, Pangkalanbaru 33684, Indonesia; 7Eijkman Research Center for Molecular Biology, National Research and Innovation Agency, Bogor 16680, Indonesia

**Keywords:** antioxidants, extraction methods, Java cardamom fruit, total flavonoid content, total phenolic content

## Abstract

Free radicals contribute to the pathophysiology of degenerative diseases which increase mortality globally, including mortality in Indonesia. *Amomum compactum* Soland. Ex Maton fruit from the Zingiberaceae family, also known as Java cardamom, contains secondary metabolites that have high antioxidant activities. The antioxidant activity of the methanol extract of Java cardamom fruit correlates with its flavonoid and phenolic compound contents, which can be affected by different methods and durations of extraction. This study aimed to measure and compare the effects of extraction methods and durations on total flavonoid and phenolic contents (TFCs and TPCs) and subsequent antioxidant activities by the 2,2′-diphenyl-1-picrylhydrazyl (DPPH) radical, ferric reducing antioxidant power (FRAP), 2,2′-azino-bis(3-ethylbenzothiazoline-6-sulfonate) (ABTS), and cupric ion reducing antioxidant capacity (CUPRAC) assays. Methanol extracts of Java cardamom were produced by continuous shaking (CSE), microwave-assisted (MAE), or ultrasonic-assisted extractions (UAE) for three different durations. CSE for 360 min resulted in the highest TFCs (3.202 mg Quercetin Equivalent/g dry weight), while the highest TPCs (1.263 mg Gallic Acid Equivalent/g dry weight) were obtained by MAE for 3 min. Out of the investigated methods, MAE for 3 min resulted in the highest antioxidant activity results for the extracts. We conclude that the polyphenolic antioxidant yield of Java cardamom depends on two parameters: the method and the duration of extraction.

## 1. Introduction

Antioxidant compounds can inhibit the activity of free radicals [1]. The human body produces endogenous or intracellular antioxidants as a natural defense mechanism against reactive oxygen species (ROS) [2]. Besides endogenous antioxidants, there are also exogenous antioxidants, such as vitamin C, vitamin E, and flavonoids, which can be obtained from fruits, tea, leaves, vegetables, and spices [2,3,4]. Due to the risk of genotoxicity and carcinogenicity, plant-based antioxidants were found to be more effective than synthetic antioxidants in decreasing ROS levels [5]. Polyphenols, such as phenolic and flavonoid compounds, are secondary plant metabolites defined as molecules with more than one phenolic ring [6]. Polyphenols are natural antioxidants that are abundant in spices, herbs, fruits, vegetables, and cereals [7]. They act as hydrogen atom donors, reducing agents, and singlet oxygen scavengers. Some polyphenols can also effectively chelate transition metals [8].

Java cardamom (*A. compactum* Sol. Ex Maton) is a spice popularly cultivated in Southeast Asia and South China and is frequently used as a traditional medicine, scent, and cooking spice. Two species of cardamom fruits, true cardamom (*Elettaria cardamomum*) and Java cardamom, can be found in Indonesia. However, due to soil factors and climatic conditions, Java cardamom is more frequently cultivated than true cardamom [9,10]. Several studies have reported on the phytochemical compounds of Java cardamom. The essential oil contents of Java cardamom range from 3.30–4.52% for seeds and 0.99–1.08% for leaves [11]. Cineole is the major phytochemical component of Java cardamom, constituting 60–80% of the volatile oil. Other components can also be found, including α- and β-pinene, camphene, limonene, α-terpineol, sabinene, terpinene, and α-humulene [10,12,13,14]. Java cardamom fruit contains various phytochemical compounds, such as alkaloids, tannins, polyphenols, saponin, flavonoids, and triterpenoids [15,16]. Contents of phenolics with hydroxyl groups, such as β-carotene and lutein [17,18], and of flavonoids in Java cardamom are considerably high [19]. High contents of bioactive compounds allow Java cardamom to have a broad range of pharmacological activities, such as antioxidant, antifungal, antibacterial, anticancer, immunomodulatory, anti-inflammatory, and anti-asthmatic effects [14].

As aforementioned, antioxidant activity is one of the pharmacological benefits of Java cardamom. Antioxidant activity can be measured by spectrophotometry by the following routinely used assays: cupric ion reducing antioxidant capacity (CUPRAC), 2,2′-azino-bis(3-ethylbenzothiazoline-6-sulfonate) (ABTS), ferric reducing antioxidant power (FRAP), 2,2′-diphenyl-1-picrylhydrazyl (DPPH), oxygen radical absorbance capacity (ORAC), and others [20]. Generally, these methods can be classified into two groups: radical scavenging methods, such as ABTS and DPPH, and non-radical redox-potential-based methods, such as FRAP and CUPRAC [21].

The methods most frequently used to extract flavonoid and phenolic compounds from spices are ultrasonication [19], maceration [18], supercritical fluid extraction [22], and Soxhlet extraction [23,24]. The method applied and the duration of extraction are the major factors that determine the amounts of antioxidants extracted [25]. Besides these, other factors, such as the polarity of the solvent [26], temperature [27], and the ratio of the sample to the solvent [28], can affect the extraction yields of compounds, which can be quantified as total flavonoid and phenolic contents (TFCs and TPCs), respectively. In this study, we aimed to evaluate and compare TFCs and TPCs, which are strongly correlated with antioxidant potential, of methanol extracts of Java cardamom fruit with regard to various durations and different methods of extraction. Our results indicated that the extraction methods applied and their durations had significant effects on the TPCs and TFCs, as well as the antioxidant activities, of the cardamom fruit extracts examined.

## 2. Results

### 2.1. Effect of Methods and Durations of Extraction on TFCs and TPCs of A. compactum Extracts

The data summarized in Table 1 show that both TFCs and TPCs were affected by different methods and durations of extraction. First, the extraction method applied significantly affected the TFCs of the extracts. A longer duration of extraction significantly increased TFCs in the case of continuous shaking extraction (CSE) and microwave-assisted extraction (MAE) methods; however, there were no significant differences between the TFC values obtained with ultrasonic-assisted extraction (UAE) across all durations. The highest total flavonoid yield (3.202 mg of quercetin equivalent (QE) per g of dry weight (DW)) was detected when the CSE method was applied with 360 min of exposure. The second-highest TFC (3.187 mg of QE per g of DW) was obtained using the MAE method with an exposure of 3 min. Contrarily, the lowest yields (1.611 and 1.661 mg of QE per g of DW) were detected when UAE was carried out for 40 min and 60 min, respectively.

In accordance with TFC values, the extraction method applied significantly affected the TPCs of the extracts as well. When CSE or MAE were carried out, longer extraction significantly increased TPCs. The duration of UAE did not affect the TPCs of the extracts. The highest yield of TPC (1.263 mg of gallic acid (GA) equivalent (GAE) per g of DW) was detected in the sample extracted by the MAE method with 3 min of exposure. In contrast, the lowest TPC (0.6722 mg of GAE per g of DW) was obtained in the sample extracted by the CSE method with 30 min of exposure. In summary, out of the investigated extraction methods and durations, CSE for 360 min and MAE for 3 min resulted in the highest TFCs and TPCs of methanol extracts of *A. compactum* (Table 1).

### 2.2. Analysis of Antioxidant Activity of A. compactum Extracts

Due to the continuous production of free radicals in the human body, which can contribute to the progression of degenerative diseases [29], the evaluation of antioxidant activities of natural extracts has gained significantly in importance. In this study, the antioxidant activities of the Java cardamom extracts that were obtained by different methods were assessed by four independent in vitro assays, namely, the ABTS, CUPRAC, FRAP, and DPPH assays (Table 2). 

On the one hand, the results observed depended significantly on the applied assay. We could detect the highest antioxidant activity (38.147 µmol of Trolox equivalent (TE) per g of DW) in methanol extracts of Java cardamom by the CUPRAC assay. In contrast, the DPPH assay resulted in the detection of the lowest antioxidant activity (3.537 µmol of TE per g of DW). When the results were analyzed by analysis of variance (ANOVA) with Duncan’s multiple range test, the data obtained by the ABTS and FRAP assays were also significantly different (*p* < 0.05) from those obtained by the CUPRAC and DPPH methods.

On the other hand, the extraction methods applied and their durations significantly affected the antioxidant activities of the extracts. The results in Table 2 show that the MAE method with 3 min of exposure resulted in the highest yields of antioxidants detected by each assay. In the case of MAE, prolonged extraction significantly increased the antioxidant activity measured by each assay. The application of the CSE method for 360 min also resulted in high antioxidant activity. Except for the results of the FRAP assay, the antioxidant activities of the extracts were increased by longer exposure to CSE. The UAE method with 20 min of exposure resulted in the highest antioxidant activity when it was measured by the ABTS and CUPRAC assays; however, the highest antioxidant activity, as determined by DPPH and FRAP assays, was obtained by 60 min and 40 min of extraction, respectively. Based on the results of the ABTS and CUPRAC assays, the duration of UAE affects the antioxidant activities of extracts. However, no significant effect of the length of UAE was observed when the DPPH and FRAP assays were carried out. In summary, out of the investigated extraction methods and durations, MAE for 3 min and CSE for 360 min resulted in the highest antioxidant activities of the methanol extracts of *A. compactum* (Table 2).

### 2.3. Antioxidant Activities of A. compactum Extracts Correlate with Their TFCs and TPCs

As a final step, a Pearson’s correlation analysis was carried out to evaluate the association between the TPCs and TFCs and the antioxidant activities of Java cardamom methanol extracts. In the case of the MAE method, which resulted in the highest antioxidant activity determined by each of the applied assays (Table 2), strong and statistically significant (*p* < 0.005) correlations were found between the TFCs and antioxidant activities of the extracts measured by the ABTS (R = 0.953) and DPPH (R = 0.875) assays. However, only statistically not significant trends of positive correlation were found between the TFCs and the antioxidant activities determined by the CUPRAC (R = 0.526) and FRAP (R = 0.875) assays (Figure 1a). TPCs and antioxidant activities measured by the ABTS assay correlated most strongly (R = 0.896, *p* = 0.001), while statistically significant (*p* < 0.05) correlations were also found when antioxidant activities were assayed by DPPH (R = 0.716) and FRAP (R = 0.712). The antioxidant activity results of the CUPRAC assay tended to be correlated (R = 0.618, *p* = 0.07) with the TPCs of the extracts (Figure 1b).

In the case of the extracts generated by CSE, statistically significant (*p* < 0.05) correlations were found between their TFCs and antioxidant activities measured by three independent assays, CUPRAC (R = 0.885), ABTS (R = 0.847), and DPPH (R = 0.714). There were no significant correlations between the TFCs and antioxidant activities of the extracts determined by the FRAP assay (R = 0.462, *p* = 0.21) (Figure 2a). TPCs and antioxidant activities measured by the CUPRAC assay correlated significantly (R = 0.699, *p* = 0.04), while the antioxidant activities measured by the other three assays did not significantly correlate with the TPCs of the extracts (Figure 2b).

The TFCs and antioxidant activities of the extracts developed by the UAE method showed a statistically significant (*p* < 0.05) positive correlation (R = 0.734) only in the case when antioxidant activity was measured by the DPPH assay (Figure 3a). In the case of the extracts generated by UAE, there were no significant correlations between the TPCs and antioxidant activities determined by any of the applied assays. The strongest trend of positive correlation (R = 0.504, *p* = 0.17) was observed when antioxidant activity was measured by the CUPRAC assay (Figure 3b).

## 3. Discussion

Polyphenols are natural compounds that are primarily synthesized by plants and possess chemical features related to phenolic substances [30]. Flavonoid and phenolic acids are two major classes of polyphenols, and both have a wide range of biological activities and play an important role in combating various diseases [31]. These compounds can be extracted from plants by using several methods with different durations. The selection of the method to be applied is critical in determining the amounts of the polyphenol compounds to be obtained, which can be quantified as the TFCs and TPCs of the extracts [25]. Not only the method type but also its duration strongly affects the efficiency and the optimal design of the extraction with respect to minimizing the energy cost of the process [32]. In our study, we have compared the effectivity of three distinct methods, MAE, CSE, and UAE, each of which was applied for three different durations, in yielding TFCs and TPCs from methanol extracts of Java cardamom fruit.

Our results showed that the greatest TPC yield was obtained using the MAE method for three minutes of exposure as compared to the CSE or UAE methods (Table 1). In accordance with our results, Upadhya et al. [25] observed that MAE resulted in the highest TPCs from methanol extracts of *Achyranthes aspera* leaves. High TPCs were also obtained when methanol extracts of brown algae species were produced by MAE [33]. The high efficiency of this method may be due to the application of a heating process without the generation of a thermal gradient and because the phenolic compounds strongly absorb microwave energy [7].

In our experiments, out of the investigated methanol extracts of Java cardamom fruit, CSE for 360 min resulted in the greatest TFC yield. We observed that the increase in the duration of CSE significantly affected the TFCs of the extracts (Table 1). Another study investigated the optimization of extraction from dried chokeberry and found that TFC positively correlated with the duration of CSE [8]. In the case of UAE from black chokeberry, the extraction yields were rapidly elevated in the first 15 min and then slowly increased further over the next 4 h [34]. When ethanol extracts of *Terminalia catappa* L. leaves were obtained by UAE, increasing TFCs and TPCs were observed with durations of 20 and 40 min. The highest TPCs were obtained from extracts produced by the UAE method with an extraction duration of 40 min. TFCs and TPCs decreased when the duration of extraction was prolonged to 60 min [35]. A sonication step with a long duration during the extraction process using the UAE method can lead to decreased diffusion rates, reduced diffusion areas, and the elevation of diffusion distances, resulting in reduced polyphenol levels [35].

We investigated the antioxidant activity of cardamom fruit extracts by using different assays, such as ABTS, DPPH, CUPRAC, and FRAP. The components of the extracts, the solvents used during the extraction process, and the characteristics of the antioxidants, such as their hydrophilicity and hydrophobicity, significantly influence the antioxidant activities of extracts [36]. Therefore, it is necessary to use various independent methods to evaluate the antioxidant activities of plant extracts, reflecting their capabilities to inhibit the negative effects of free radicals [37]. ABTS and DPPH assays were used to evaluate the free radical scavenging activities of cardamom fruit extracts [38], whereas FRAP and CUPRAC assays were carried out to investigate their reducing power activities [39]. The quantification of antioxidant activity in the case of each obtained method is referenced based on the Trolox standard curve.

Our results showed that the reducing powers of the extracts were greater when they were determined by the CUPRAC instead of the FRAP assay. The reducing power assay is commonly used to elucidate the potential of an antioxidant to donate an electron, which is an important biological feature of phenolic antioxidant compounds [40]. The CUPRAC assay is based on the reducing power antioxidant activity that converts cupric (Cu^2+^) to cuprous (Cu^+^) ions in parallel with the production of a chromophore, with maximum absorption measured at 450 nm [12]. In association with the results of TPC measurements yielded by each method (Table 1), the highest antioxidant activity was obtained by the MAE method for three minutes of exposure (Table 2). MAE is an extraction method that uses micro radiation, which results in overheating in cells, leading to cell wall damage and increasing the TPCs released from cell matrices [7].

The radical scavenging activities of the extracts were evaluated by ABTS and DPPH assays. We found that higher antioxidant activities were detected when they were measured by the ABTS as compared to the DPPH assay. In association with the results of reducing power measurements, the highest radical scavenging activities were also observed in the samples that were extracted using MAE with three minutes of exposure (Table 2). The ABTS assay, also known as the TE antioxidant capacity assay, was developed based on the interaction between antioxidants and ABTS radical cations [41] that subsequently form a blue-green color, with the maximal absorption at wavelengths of 414, 645, 734, and 815 nm [42]. The optimal wavelength for the measurement is 734 nm because of the possible interruptions from other compounds that absorb light at other wavelengths [43]. The intensity of the chromophore, which is generated by oxidation, decreases as antioxidant activity increases [42]. It was reported previously and reproduced by our data that the higher TFC and TPC yielded by an optimized method and duration of extraction positively correlated with the antioxidant activity of the extract in question [25].

The lowest antioxidant activity of the investigated cardamom fruit methanol extracts was obtained by the CSE method with 30 min of exposure and quantified by the DPPH assay (Table 2). DPPH is one of the most accurate and frequently used assays for the evaluation of antioxidant activity [44]. The principle of this assay is based on the donation of H^+^ to DPPH radicals, which corresponds to a color alteration from violet to pale yellow in the solution [45]. As reported previously, temperature and duration of CSE can influence the extraction yield and the antioxidant activity obtained. A low temperature and short duration of extraction may result in low antioxidant activity because of a lack of interaction between the solvent and the sample [25].

The correlation between TPC and TFC along with antioxidant activity with respect to each method and duration of extraction was analyzed by Pearson’s correlation analysis [46,47]. We found that the TFCs of the samples extracted by CSE positively correlated with their antioxidant activities measured by the ABTS, CUPRAC, and DPPH assays (Figure 2a). Samples extracted by the MAE method showed a positive correlation between their TPCs and antioxidant activities which was statistically significant for all of the applied methods, except for the CUPRAC assay (Figure 1b). A significant positive correlation was found between the TPCs and antioxidant activities, measured by the CUPRAC assay, of the samples extracted by CSE (Figure 2b). In contrast, no significant correlation was observed between the TPCs and antioxidant activities of the samples produced by the UAE method (Figure 3b). The extracted antioxidant compounds can potentially suppress inflammation, protect endothelial cell membranes, and subsequently prevent cellular damage [48]. We reported previously that cardamom fruit extracts possessed high flavonoid contents and antioxidant activities depending on their regional origins [19]. To date, the greatest flavonoid contents and antioxidant activities for cardamom fruit have been obtained by aqueous extracts [49]. Solvent–sample interaction strongly affects flavonoid content because the solution efficiency determines the structure and polarity of the compounds obtained from a sample matrix [50].

Natural antioxidant compounds have attracted the attention of researchers for decades because of the rarity of their side effects and low toxicities [51]. Cardamom fruits have various pharmacological activities, such as anti-inflammatory [52], anti-atherosclerotic [53], antibacterial [54], and anti-cancer effects [55]. An in vivo study reported that ethanol extract of Java cardamom decreased levels of ROS and T helper (Th) 2 cytokines, including interleukin (IL)-4 and -5, in the bronchoalveolar lavage fluid of ovalbumin-induced asthmatic mice [56]. Another in vivo study elucidated the anti-inflammatory activity of Java cardamom ethanol extract in lipopolysaccharide-treated mice. Java cardamom ethanol extract strongly inhibited the generation of nitric oxide (NO), prostaglandin E2 (PGE2), IL-6, and tumor necrosis factor (TNF)-α, and inhibited the protein expression of inducible NO synthase and cyclooxygenase-2 [57]. Further research is needed to find the molecular targets of the biologically active compounds within the extracts which underlie the beneficial effects of this traditional medicinal product. 

Our study has demonstrated that various methods and durations of extraction influence the TPCs, TFCs, and antioxidant activities of cardamom fruit methanol extracts; therefore, the proper methods and durations of extraction should be optimized in future studies for the continuous effective production of Java cardamom-based antioxidant products. Since Indonesia has a great potential to cultivate Java cardamom, studies on the optimization of its extraction can help to create a basis for the systematic production of this pharmaceutical in the Southeast Asian region [58].

## 4. Materials and Methods

### 4.1. Chemicals

Methanol (pro-analysis), Folin–Ciocalteu phenol reagent, aquadest, aquabidest, GA, ammonium acetate buffer, CuCl_2_, K_2_S_2_O_4_, ammonium acetate buffer, neocuproine, AlCl_3_, FeCl_3_, HCl, and quercetin were obtained from Merck-Millipore (Darmstadt, Germany). Trolox, ABTS, sodium carbonate, glacial acetate acid, and DPPH were obtained from Sigma-Aldrich (St. Louis, MO, USA). 2,4,6-tripydyl-s-triazine (TPTZ) and acetic acid were obtained from Sisco Research Laboratories Pvt. Ltd. (Maharashtra, India).

### 4.2. Sample Preparation

Java cardamom fruit was obtained from Tropical Biopharmaca Research Center IPB University (Bogor, Indonesia). The material was dried at 45 °C for 48 h. The dried material was ground to yield an 80-mesh powder.

### 4.3. CSE

Methanol extract of cardamom fruit was obtained by CSE that was carried out based on a previously described method [25] with modifications. Briefly, 5 g of Java cardamom fruit powder was put into a beaker to which 100 mL of pro-analytical methanol solvent was added. Then, the mixture was placed in a water bath shaker. Stirring was carried out constantly at a speed of 110 rpm at a controlled temperature (25 °C). Extractions were carried out for 30, 180, or 360 min, separately. The extract was then filtered using Whatman filter paper No. 1 and re-volumized (to obtain a 5% concentration). Extraction was carried out with 3 repetitions for individual plants.

### 4.4. MAE

Methanol extract of cardamom fruit was obtained by MAE that was carried out based on a previously described method [25] with modifications. A total of 1 g of Java cardamom fruit powder was put into an Erlenmeyer flask to which 20 mL of pro-analytical methanol solvent was added. The flasks were exposed to a microwave oven (Sharp R-21D0(S)-IN) at 135 W for 1, 2, or 3 min. The suspension was cooled periodically and then filtered using Whatman filter paper No. 1, and the final volume was set to 20 mL (to obtain a 5% concentration). Extraction was carried out with 3 repetitions for individual plants.

### 4.5. UAE

Methanol extract of cardamom fruit was obtained by UAE that was carried out based on a previously described method [32] with modifications. A total of 1 g of Java cardamom fruit powder was put into an Erlenmeyer flask to which 20 mL of pro-analytical methanol solvent was added. The flasks were covered with aluminum foil and were placed in a sonicator bath. Samples were sonicated for 20, 40, or 60 min at room temperature. The sonicated flasks were then centrifuged at 10,000× *g* and 4 °C for 15 min. The supernatant was then separated from the pellets and the extract was re-volumized (to obtain a 5% concentration). Extraction was carried out with 3 repetitions for individual plants.

### 4.6. Quantification of TFC

Quantification of TFC was carried out based on a previously described method [39] with modifications, using quercetin as a flavonoid standard. The calibration of the standard resulted in a line equation of y = 0.0022x + 0.0125, with an R^2^ value of 0.9895, and the total flavonoid levels were expressed in mg of QE per g of DW. The Java cardamom fruit extract (25 μL) was added to 120 μL distilled water, 10 μL of 10% AlCl_3_, 10 μL of glacial acetate acid, and 50 μL of methanol. The solution was incubated for 30 min and the absorbance was measured at 409 nm using a nano-spectrophotometer (SPECTROstarNano BMG LABTECH, Offenburg, Germany). TFC was expressed as mg of QE per g of DW, with quercetin variation standard 0–500 ppm.

### 4.7. Quantification of TPC

TPC was quantified based on a previously described method [59] with modifications, using Folin–Ciocalteau phenol reagent with GA as a standard. A quantity of 20 µL of methanol extract of Java cardamom fruit was mixed with 120 µL of 10% Folin–Ciocalteu phenol reagent in a microplate. The mixture was allowed to react for 5 min in a dark room and then 80 µL of 10% sodium carbonate solution was added. The mixture was incubated for 2 h at room temperature until the color was formed. The absorbance of the solution was read at 750 nm using a nano-spectrophotometer (SPECTROstarNano BMG LABTECH). TPC was calculated based on the GA standard and expressed as mg of GAE per g of DW.

### 4.8. Quantification of Antioxidant Activities

The following assays were carried out based on previously described methods [28] with modifications. Trolox was used as standard in the aforementioned assays, and the results were expressed as µmol of TE per g of DW or TE antioxidant capacity.

#### 4.8.1. ABTS Assay

A quantity of 20 µL of the extracted sample was added to 180 µL of ABTS reagent on a microplate. Then, the mixture was incubated at 30 °C for 6 min. The absorbance of the solution was measured at a wavelength of 734 nm using a nano-spectrophotometer (SPECTROstarNano BMG LABTECH). ABTS activity was calculated based on the Trolox standard and expressed as µmol of TE per g of DW.

#### 4.8.2. CUPRAC Assay

The extracted sample (50 µL) was added to the following mixture: 50 µL of 0.01 M CuCl_2_, 50 µL of ammonium acetate buffer (pH 7), 50 µL of 0.0075 M neocuproine, in a 96-well microplate. Then, the mixture was homogenized with a vortex for 10 s and incubated for 30 min in dark room. The absorbance of the solution was measured at a wavelength of 450 nm using a nano-spectrophotometer (SPECTROstarNano BMG LABTECH). CUPRAC activity was calculated based on the Trolox standard and expressed as µmol of TE per g of DW.

#### 4.8.3. DPPH Assay

A quantity of 100 µL of the extracted sample was added to 100 μL 125 μM DPPH. The solution was incubated in a test tube at 37 °C for 30 min. The absorbance was measured with a nano-spectrophotometer (SPECTROstarNano BMG LABTECH) at 515 nm. Antioxidant activity was indicated by a change in color from dark purple to yellow. Free radical scavenging activity was expressed in μmol of TE per g of DW, with Trolox standard of 0–50 μM.

#### 4.8.4. FRAP Assay

FRAP solution was prepared by mixing acetate buffer with a pH of 3.6, 10 μM TPTZ dissolved in 40 mM HCl, and 20 mM FeCl_3_ solution in a ratio of 10:1:1, which was protected from light. Then, 300 μL FRAP solution was added to a 10 μL sample. The mixture was vortexed and incubated at 37 °C for 5 min. After incubation, the absorbance was recorded at 593 nm with a nano-spectrophotometer (SPECTROstarNano BMG LABTECH). The results of the FRAP assay were expressed as μmol of TE per g of DW, with Trolox standard of 0–400 μM.

### 4.9. Statistics and Figure Preparation

All measured values are expressed as means ± standard deviations (SDs) for the number of independent repetitions indicated. Statistical analysis was carried out using one-way ANOVA with Duncan’s multiple range test. Pearson’s correlations were calculated using SPSS v. 22 (IBM, Armonk, NY, USA). GraphPad Prism 9 (GraphPad Software, San Diego, CA, USA) was used for figure preparation.

## 5. Conclusions

Different methods and durations of extraction significantly affected the TFCs, TPCs, and antioxidant activities of methanol extracts of Java cardamom fruit. The highest total phenolic and flavonoid contents were obtained when extraction was performed by the CSE method with a duration of 360 min and by the MAE method with a duration of 3 min. No difference was observed when different durations were applied for the UAE method. Antioxidant activity, which was measured by CUPRAC, FRAP, DPPH, and ABTS, positively correlated with TFC and TPC when the extraction was performed by MAE. Positive correlations between antioxidant activity and TFC and TPC were only observed for measurements by CUPRAC when the extraction was performed by CSE. Extraction performed by UAE did not show any significant correlations between antioxidant activity and TFC and TPC. 

## Figures and Tables

**Figure 1 plants-11-02221-f001:**
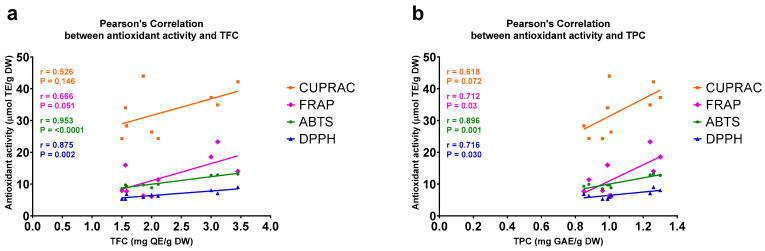
Scatter plot displaying Pearson’s correlations between antioxidant activities and (**a**) total flavonoid contents (TFCs) and (**b**) total phenolic contents (TPCs) of the methanol extracts of *A. compactum* that were generated by microwave-assisted extraction (MAE). The antioxidant activities of the extracts were measured by four independent assays. ABTS, 2,2′-azino-bis(3-ethylbenzothiazoline-6-sulfonate); CUPRAC, cupric ion reducing antioxidant capacity; DPPH, 2,2′-diphenyl-1-picrylhydrazyl; DW, dry weight; FRAP, ferric reducing antioxidant power; GAE, gallic acid equivalent; QE, quercetin equivalent; TE, Trolox equivalent.

**Figure 2 plants-11-02221-f002:**
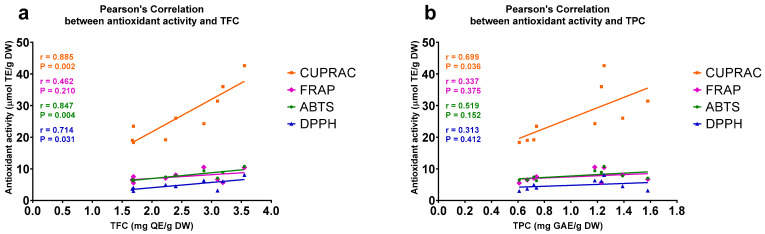
Scatter plot displaying Pearson’s correlations between antioxidant activities and (**a**) total flavonoid contents (TFCs) and (**b**) total phenolic contents (TPCs) of methanol extracts of *A. compactum* that were generated by continuous shaking extraction (CSE). The antioxidant activities of the extracts were measured by four independent assays. ABTS, 2,2′-azino-bis(3-ethylbenzothiazoline-6-sulfonate); CUPRAC, cupric ion reducing antioxidant capacity; DPPH, 2,2′-diphenyl-1-picrylhydrazyl; DW, dry weight; FRAP, ferric reducing antioxidant power; GAE, gallic acid equivalent; QE, quercetin equivalent; TE, Trolox equivalent.

**Figure 3 plants-11-02221-f003:**
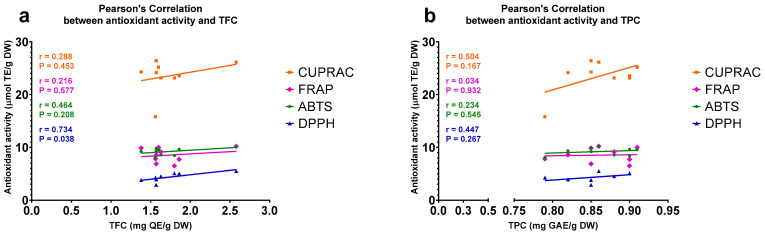
Scatter plot displaying Pearson’s correlations between antioxidant activities and (**a**) total flavonoid contents (TFCs) and (**b**) total phenolic contents (TPCs) of methanol extracts of *A. compactum* that were generated by ultrasonic-assisted extraction (UAE). The antioxidant activities of the extracts were measured by four independent assays. ABTS, 2,2′-azino-bis(3-ethylbenzothiazoline-6-sulfonate); CUPRAC, cupric ion reducing antioxidant capacity; DPPH, 2,2′-diphenyl-1-picrylhydrazyl; DW, dry weight; FRAP, ferric reducing antioxidant power; GAE, gallic acid equivalent; QE, quercetin equivalent; TE, Trolox equivalent.

**Table 1 plants-11-02221-t001:** Total flavonoid and phenolic contents (TFCs and TPCs) of methanol extracts of *A. compactum* obtained by different extraction methods.

Extraction Method	Duration (min)	TFC (mg QE g^−1^ DW)	TPC (mg GAE g^−1^ DW)
CSE	30	1.684 ± 0.010 ^c^	0.672 ± 0.063 ^c^
	180	2.575 ± 0.459 ^b^	1.233 ± 0.451 ^a^
	360	3.202 ± 0.337 ^a^	1.221 ± 0.034 ^a^
MAE	1	1.729 ± 0.333 ^c^	0.895 ± 0.054 ^b,c^
	2	1.809 ± 0.228 ^c^	1.000 ± 0.008 ^a,b^
	3	3.187 ± 0.232 ^a^	1.263 ± 0.029 ^a^
UAE	20	1.904 ± 0.582 ^c^	0.841 ± 0.021 ^b,c^
	40	1.611 ± 0.240 ^c^	0.889 ± 0.031 ^b,c^
	60	1.661 ± 0.123 ^c^	0.858 ± 0.057 ^b,c^

Data are presented as means ± SDs. Different letter notations indicate that there was a significant difference between the groups analyzed by ANOVA with Duncan’s multiple range test at a 95% confidence level. CSE, continuous shaking extraction; DW, dry weight; GAE, gallic acid equivalent; MAE, microwave-assisted extraction; QE, quercetin equivalent; UAE, ultrasonic-assisted extraction.

**Table 2 plants-11-02221-t002:** Antioxidant activities of methanol extracts of *A. compactum* measured by different methods.

Extraction Method	Duration (min)	ABTS (µmol TE g^−1^ DW)	DPPH (µmol TE g^−1^ DW)	CUPRAC (µmol TE g^−1^ DW)	FRAP (µmol TE g^−1^ DW)
CSE	30	6.380 ± 0.114 ^e^	3.537 ± 0.502 ^e^	20.280 ± 2.791 ^c^	6.497 ± 1.000 ^b^
	180	7.409 ± 0.384 ^d^	4.158 ± 0.936 ^e^	25.547 ± 6.113 ^b,c^	7.251 ± 0.707 ^b^
	360	9.683 ± 0.974 ^b^	6.187 ± 1.081 ^a,b^	34.302 ± 9.286 ^a,b^	8.848 ± 2.796 ^b^
MAE	1	9.263 ± 0.703 ^b^	6.177 ± 0.794 ^b,c^	25.658 ± 2.329 ^b,c^	9.041 ± 2.052 ^b^
	2	9.441 ± 0.515 ^b^	5.982 ± 0.669 ^b,c,d^	34.791 ± 8.859 ^a,b^	9.462 ± 5.652 ^b^
	3	12.933 ± 0.209 ^a^	8.078 ± 0.960 ^a^	38.147 ± 3.710 ^a^	18.637± 4.631 ^a^
UAE	20	9.772 ± 0.414 ^b^	4.100 ± 1.318 ^e^	25.569 ± 1.239 ^b,c^	8.549 ± 1.658 ^b^
	40	9.478 ± 0.230 ^b^	4.420 ± 0.864 ^d,e^	24.347 ± 0.835 ^c^	9.198 ± 1.277 ^b^
	60	8.379 ± 0.241 ^c^	4.624 ± 0.419 ^c,d,e^	20.702 ± 4.234 ^c^	7.830 ± 1.315 ^b^

Data are presented as means ± SDs. Different letter notations indicate that there was a significant difference between the groups analyzed by ANOVA with Duncan’s multiple range test at a 95% confidence level. ABTS, 2,2′-azino-bis(3-ethylbenzothiazoline-6-sulfonate); CSE, continuous shaking extraction; CUPRAC, cupric ion reducing antioxidant capacity; DPPH, 2,2′-diphenyl-1-picrylhydrazyl; DW, dry weight; FRAP, ferric reducing antioxidant power; MAE, microwave-assisted extraction; TE, Trolox equivalent; UAE, ultrasonic-assisted extraction.

## Data Availability

The data presented in this study are available within the article.

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
