# Peer review of "Effects of Methods and Durations of Extraction on Total Flavonoid and Phenolic Contents and Antioxidant Activity of Java Cardamom (Amomum compactum Soland Ex Maton) Fruit"

_plants, 2022, doi:10.3390/plants11172221_

Round 1

Reviewer 1 Report

Even though this analysis is done for the first time to this plant species, the results are more than expected and obvious. The novelty of this research is very limited to the plant species. Such studies were done numerous times in past and for many plant materials. Please see:

https://www.sciencedirect.com/science/article/abs/pii/S0260877406006649

Anyhow, since it is first study of this interest to this important spice, and since the article is well written, there are good reasons to accept it.

Author Response

We would like to thank Reviewer 1 for the positive and constructive review. Traditional medicine made from various natural herbs, which is popularly called jamu, has been a preserved culture in Indonesia to cure various diseases which underlies the importance of the current study. We have cited the article suggested by Reviewer 1 as ref. [32] in our revised manuscript.

Reviewer 2 Report

This paper deals with the impact of extraction method on the antioxidant activity of Java cardamom fruits extracts. Among the main issues of the paper is the complete lack of discussion of the extract composition, besides authors rely only on the total flavonoid and polyphenol contents.

Some other points are as follows:

Abstract. The author names of the plants should be introduced at the first mention along with the family name. The author name of the species should be embedded into the title.

The same for the abbreviations for extraction methods.

Please give a little more details regarding the beneficial effects of java cardamom. Recently, “Amomum compactum: review on pharmacological studies” has been published: Plant Cell Biotechnology and Molecular Biology 22(33&34):61-69; 2021. In addition, antioxidant effects of leaves, seeds and rhizomes have been evaluated as well.

l. 58-61 should be rephrased (...saponin?...tanin?). The cited references don’t provide the data on the java cardamon phytochemicals. Please, provide appropriate data.

The study’s aim is not justified in a convincing manner. Why the methanol was chosen for the extraction. Solvent type is also essential in the extraction method.

Some data from the results should be moved to the Experimental part” l. 90-92, 102-103, 127-129.

Please, provide the extraction yield for each extract.

The obtained results should be related to those of previous antioxidant studies of java cardamom (Widowati et al. Biomed Eng, 2015; 1: 24; Pujiarti et al, Wood Res J, 2020; 11:35; Nurcholis et al. Ann Agric Sci, 2021; 66: 58).

l. 216 should be rephrased: there are different classes/subclasses of polyphenols (not varieties). Both MAE and UAE could be discussed in terms of the green chemistry methods. The repetition in the Results and Discussion sections should be avoided.

All results have been drawn based on one sample batch. How many plant individuals were used for the experiments? No independent extracts have been made from each of type of sample or from different sample batches, so generalizing just from a single extract can be very risky.

l. 337 pro-analytical methanol? (methanol (pro analysis) should be mentioned in chemicals).

Is it controlled the temperature with this standard deviation 25±5º C? (RSD =20%)

l. 362 aquadest?

Author Response

This paper deals with the impact of extraction method on the antioxidant activity of Java cardamom fruits extracts. Among the main issues of the paper is the complete lack of discussion of the extract composition, besides authors rely only on the total flavonoid and polyphenol contents.

Some other points are as follows:

Abstract. The author names of the plants should be introduced at the first mention along with the family name. The author name of the species should be embedded into the title. The same for the abbreviations for extraction methods.

Author’s Response:

Thank you for the suggestion. The author names of the plants and abbreviations were corrected.

Please give a little more details regarding the beneficial effects of java cardamom. Recently, “Amomum compactum: review on pharmacological studies” has been published: Plant Cell Biotechnology and Molecular Biology 22(33&34):61-69; 2021. In addition, antioxidant effects of leaves, seeds and rhizomes have been evaluated as well.

l. 58-61 should be rephrased (...saponin?...tanin?). The cited references don’t provide the data on the java cardamon phytochemicals. Please, provide appropriate data.

Author’s Response:

Thank you for the suggestion. We have added more information about the beneficial effects of A. compactum (lanes 63-73) and cited correct references (10-19) for the phytochemical content of A. compactum. Saponin and tannin are phytochemical agents found in the investigated plant.

The study’s aim is not justified in a convincing manner. Why the methanol was chosen for the extraction. Solvent type is also essential in the extraction method.

Author’s Response:

Thank you for the input provided. We agree that apart from the method, the choice of solvent is also important in the optimization of the extraction. The choice of methanol solvent was aimed at exploring more deeply the methanol extract of the studied plant. Due to the use of other solvents such as ethanol and ethyl acetate we have done, and we have published in ref 19. Thus, the objective is to emphasize the extraction method with the selected solvent methanol.

Some data from the results should be moved to the Experimental part” l. 90-92, 102-103, 127-129.

Author’s Response:

We have amended these parts.

Please, provide the extraction yield for each extract.

Author’s Response:

Thank you for the input provided. In this study, we did not determine the extraction yield. This study determined the content of TPC, TFC, and antioxidant activity based on the dry weight of the sample. In this case, the extract concentration was 0.05 g/mL dry weight. This approach has been widely carried out by several studies that have been published previously, such as in the following article https://doi.org/10.1016/j.sajb.2018.10.026 and ref 19.

The obtained results should be related to those of previous antioxidant studies of java cardamom (Widowati et al. Biomed Eng, 2015; 1: 24; Pujiarti et al, Wood Res J, 2020; 11:35; Nurcholis et al. Ann Agric Sci, 2021; 66: 58).

l. 216 should be rephrased: there are different classes/subclasses of polyphenols (not varieties). Both MAE and UAE could be discussed in terms of the green chemistry methods. The repetition in the Results and Discussion sections should be avoided.

Author’s Response:

Thank you for the constructive input. We have added more explanation and rephrased the text according to the Reviewer’s suggestions (lanes 217-220).

All results have been drawn based on one sample batch. How many plant individuals were used for the experiments? No independent extracts have been made from each of type of sample or from different sample batches, so generalizing just from a single extract can be very risky.

Author’s Response:

Thank you for the input given. The extraction process was carried out with 3 repetitions of individual plants. This explanation has been added to the text (lanes 353-354, 362-363, 372-373).

l. 337 pro-analytical methanol? (methanol (pro analysis) should be mentioned in chemicals).

Author’s Response:

We apologize for the mistake. Thank you!

Is it controlled the temperature with this standard deviation 25±5º C? (RSD =20%)

Author’s Response:

We apologize for the mistake. Thank you!

l. 362 aquadest?

Author’s Response:

Corrected. Thank you!

Reviewer 3 Report

The study is of a methodological nature and is intended to determine the most efficient method of extraction of flavonoids and phenolics from cardamom fruit and to evaluate and compare the antioxidant activity of the extracts obtained by different methods. Such works are very necessary and relevant.

I think that the aim of the paper is not well formulated and the last sentence of the introduction already belongs to the conclusions section.

Lines 86–87; 120–121; 160: The chapter titles are too long and the writing style is not appropriate, as it would be for a conclusion.

The title of the first table lacks the indication that the methanol extracts were prepared by a different method.

Line 56: the name of the species should be written in italic font

Lines 63-64:  The sentence should be rewritten: Another source of antioxidants in cardamom is bioactive phenolic compounds with hydroxyl groups such as β-carotene and lutein [10].

Lines 122-123: Sentences should belong in the discussion section, not in the results section.

Lines 127-128: Sentences should belong in the Materials and Methods section

I think the conclusions should be more detailed and clearer. Some of them are not clear at all. For example, what should be meant by the conclusion: "When the MAE method was applied for three minutes, the highest antioxidant activity was detected by ABTS, CUPRAC, DPPH, or FRAP assays, respectively", whether it should be a decrease in antioxidant activity or the same for all detection methods.

Author Response

The study is of a methodological nature and is intended to determine the most efficient method of extraction of flavonoids and phenolics from cardamom fruit and to evaluate and compare the antioxidant activity of the extracts obtained by different methods. Such works are very necessary and relevant.

I think that the aim of the paper is not well formulated and the last sentence of the introduction already belongs to the conclusions section.

Lines 86–87; 120–121; 160: The chapter titles are too long and the writing style is not appropriate, as it would be for a conclusion.

The title of the first table lacks the indication that the methanol extracts were prepared by a different method.

Line 56: the name of the species should be written in italic font

Lines 63-64:  The sentence should be rewritten: Another source of antioxidants in cardamom is bioactive phenolic compounds with hydroxyl groups such as β-carotene and lutein [10].

Lines 122-123: Sentences should belong in the discussion section, not in the results section.

Lines 127-128: Sentences should belong in the Materials and Methods section

Author’s Response:

Thank you for the suggestions. We have corrected the manuscript following the suggestions from the Reviewer.

I think the conclusions should be more detailed and clearer. Some of them are not clear at all. For example, what should be meant by the conclusion: "When the MAE method was applied for three minutes, the highest antioxidant activity was detected by ABTS, CUPRAC, DPPH, or FRAP assays, respectively", whether it should be a decrease in antioxidant activity or the same for all detection methods.

Author’s Response:

Thank you for the suggestions. We have re-written the conclusion part to make it clearer and more detailed.

Round 2

Reviewer 2 Report

The Authors have addressed my comments from the first round. The manuscript has been improved according to all my suggestions. I have no more remarks.

Please, check the english editing.

Reviewer 3 Report

The authors have responded to all comments.